# Improvement of Left Ventricular Global Longitudinal Strain after 6-Month Therapy with GLP-1RAs Semaglutide and Dulaglutide in Type 2 Diabetes Mellitus: A Pilot Study

**DOI:** 10.3390/jcm12041586

**Published:** 2023-02-16

**Authors:** Paolo Basile, Andrea Igoren Guaricci, Giuseppina Piazzolla, Sara Volpe, Alfredo Vozza, Marina Benedetto, Maria Cristina Carella, Daniela Santoro, Francesco Monitillo, Andrea Baggiano, Saima Mushtaq, Laura Fusini, Fabio Fazzari, Cinzia Forleo, Nunziata Ribecco, Gianluca Pontone, Carlo Sabbà, Marco Matteo Ciccone

**Affiliations:** 1University Cardiology Unit, Interdisciplinary Department of Medicine, Policlinic University Hospital, 70121 Bari, Italy; 2Interdisciplinary Department of Medicine, School of Medicine, University of Bari “Aldo Moro”, 70121 Bari, Italy; 3Department of Economics and Finance, University of Bari—Aldo Moro, 70121 Bari, Italy; 4Department of Perioperative Cardiology and Cardiovascular Imaging, Centro Cardiologico Monzino, IRCCS, 20138 Milan, Italy

**Keywords:** Glucagone Like Peptide-1 Receptor Agonists, dulaglutide, semaglutide, GLP-1 RA, Diabetes Mellitus type 2, Global Longitudinal Strain, GLS, diabetic cardiomyopathy, cardiac function

## Abstract

(1) Background: Glucagone-Like Peptide-1 Receptor Agonists (GLP-1 RAs) (GLP-1 RAs) are incretine-based medications recommended in the treatment of type 2 Diabetes Mellitus (DM2) with atherosclerotic cardiovascular disease (ASCVD) or high or very high cardiovascular (CV) risk. However, knowledge of the direct mechanism of GLP-1 RAs on cardiac function is modest and not yet fully elucidated. Left ventricular (LV) Global Longitudinal Strain (GLS) with Speckle Tracking Echocardiography (STE) represents an innovative technique for the evaluation of myocardial contractility. (2) Methods: an observational, perspective, monocentric study was conducted in a cohort of 22 consecutive patients with DM2 and ASCVD or high/very high CV risk, enrolled between December 2019 and March 2020 and treated with GLP-1 RAs dulaglutide or semaglutide. The echocardiographic parameters of diastolic and systolic function were recorded at baseline and after six months of treatment. (3) Results: the mean age of the sample was 65 ± 10 years with a prevalence of the male sex (64%). A significant improvement in the LV GLS (mean difference: −1.4 ± 1.1%; *p* value < 0.001) was observed after six months of treatment with GLP-1 RAs dulaglutide or semaglutide. No relevant changes were seen in the other echocardiographic parameters. (4) Conclusions: six months of treatment with GLP-1 RAs dulaglutide or semaglutide leads to an improvement in the LV GLS in subjects with DM2 with and high/very high risk for ASCVD or with ASCVD. Further studies on larger populations and with a longer follow-up are warranted to confirm these preliminary results.

## 1. Introduction

According to the International Diabetes Federation (IDF), diabetic individuals comprise almost 9% of the entire world’s population [1]. Moreover, this percentage tends to increase over time in parallel with the average survival of the population. Over the past two decades, Diabetes Mellitus (DM) has been considered a “coronary risk equivalent” on the basis of diagnostic and prognostic studies [2,3,4,5,6,7,8]. With these assumptions, DM has been the target of maximal therapy aimed at lowering glycaemia and, at the same time, reducing cardiovascular complications.

Glucagone-Like Peptide-1 Receptor Agonists (GLP-1 RAs) (GLP-1RAs) are incretine-based glucose-lowering medications widely used in the treatment of Diabetes Mellitus type 2 (DM2) [9,10]. Beyond the effectiveness on the glycaemic control, these antidiabetic drugs demonstrated a reduction in major adverse cardiovascular events (MACE) in DM2 patients, such that recent European Society of Cardiology (ESC) guidelines strongly recommend their use [11,12,13,14,15,16,17,18]. In fact, GLP-1 RAs have demonstrated pleiotropic effects, reducing oxidative stress and inflammation, reversing cardiac remodelling and preserving renal function [19]. However, the understanding of the effects of GLP-1 RAs on the structures and function of the heart, mainly based on liraglutide and exenatide studies, is not yet fully elucidated [20]. The current analysis sought to explore the effect of GLP-1 RAs semaglutide and dulaglutide on myocardial contractility using the left ventricular (LV) Global Longitudinal Strain (GLS) Speckle Tracking Echocardiography (STE) technique. Left ventricle GLS-STE is an emerging diagnostic method for the evaluation of myocardial function [21,22]. It represents a very sensitive marker of myocardial function impairment, preceding the development of clinically relevant systolic dysfunction. Thus, its prompt recognition may allow for the prescription of treatment without delay, preventing severe cardiac complications [23]. 

## 2. Materials and Methods

### 2.1. Study Design

We conducted an observational, perspective, monocentric study in consecutive patients with DM2 to assess the effects of GLP1-RAs semaglutide and dulaglutide on cardiac function. The Inclusion criteria were: a diagnosis of DM2 and high or very high risk for cardiovascular (CV) disease or with an evident atherosclerotic cardiovascular disease (ASCVD); age ≥ 18 years; GLP-1 RAs naïve. The exclusion criteria were: contraindication for the prescription of GLP-1 Ras; previous treatment with anti-diabetic drugs or treatment with stable doses of metformin for at least 3 months; permanent atrial fibrillation or rhythm disorders; the presence of prosthetic valves; cancer cachexia; pregnancy and/or breastfeeding. Written informed consent was obtained from each participant before enrolment. The study was approved by the local ethics committee and conducted according to the principles of the revised Declaration of Helsinki.

### 2.2. Screening Protocol and Data Acquisition

A preliminary screening visit was performed at the Diabetology Clinic of the Internal Medicine Unit of our tertiary hospital to assess the eligibility of patients, according to the criteria previously provided, and to obtain the informed consent. The demographic and anthropometric data (age, sex, body mass index, body surface area, systolic and diastolic blood pressure, heart rate), the presence of CV risk factors (hypertension, dyslipidaemias, smoke habit, obesity, familiar history of CV disease), drugs assumption, comorbidity and laboratory data (renal function and serum electrolytes, hepatic function, plasma fasting glucose and glycated haemoglobin) were collected in a paper case report form. Then, the data were recorded in an electronic spreadsheet by two investigators. After the enrolment, subcutaneous dulaglutide (1.5 mg or 0.75 mg weekly) or semaglutide (0.5 or 1.0 mg weekly) was prescribed.

### 2.3. Echocardiographic Evaluation

Each patient underwent echocardiographic exams at baseline, before the initiation of GLP-1 RA, and in the follow-up, after 6 months of treatment. The echocardiographic examination was performed in a dedicated echocardiographic laboratory in the outpatient Cardiologic Clinic of the same hospital, using Canon Aplio i700 (Canon Medical System, Zoetermeer, The Netherlands) by a cardiologist trained in echocardiographic imaging (with at least 15 years of experience) and according to the recommendations stated by the American Society of Ecocardiography (ASE) and the European Society of Cardiovascular Imaging (EACVI) [24]. Before the echocardiographic evaluation, each patient was placed in the supine position for 5 min in a quiet room. Subsequently, we measured the brachial blood pressure and heart rate in the right arm with an automated digital oscillometric sphygmomanometer (A&D Medical Ltd, Tokyo, Japan). The values obtained are those included in the study and used for statistical analysis. The 2D-guided linear measurements were taken from a parasternal long axis (LV septum diameter, end-diastole LV diameter, LV posterior wall diameter, anteroposterior diameter of the left atrium) and short axis view (right ventricular outflow tract diameter) [25]. The LV mass was calculated by the linear method using the cube formula. From the apical four chamber view, the LV end diastolic volume, LV end systolic volume, left atrium volume, TAPSE (Tricuspid Annular Plane Systolic Excursion) and right ventricular inflow diameter were obtained. The left ventricular ejection fraction (LVEF) was obtained from the LV end-diastolic and end-systolic volumes using the biplane disk summation method. The right ventricular systolic function was also evaluated by the fractional area change (FAC), using the end-diastolic and end-systolic area of the right ventricle. A pulsed wave (PW) doppler at the mitral inflow and Tissue Doppler Imaging (TDI) at the mitral annular septal and lateral level in the four-chamber view were performed, then the E/A and the E/E’ ratios were calculated to assess the diastolic function of the left ventricle. A continuous wave (CW) doppler was used to assess the tricuspid regurgitation jet velocity in order to estimate the Systolic Pulmonary Artery Pressure (SPAP). Colour flow evaluation was used to define, in a semiquantitative way, the presence of valvular disease. The evaluation of the diameter of the inferior vena cava and collapse was performed in the subcostal view. The LV GLS-STE was calculated by a dedicated software from the apical four, two and three chamber view using an automated tracking algorithm to outline the myocardial borders throughout the cardiac cycle. If necessary, manual adjustments were performed to ensure the correct tracing of the endocardial border (Figure 1). A value around −20% was considered as a cut off for normality [24].

### 2.4. Study Endpoints

The endpoint of the study was to assess cardiac functional changes by echocardiography and, in particular, by GLS-STE, comparing the data at the baseline evaluation with the 6-month follow-up. Due to the explorative purpose of the study and the absence of previous data regarding the effects of dulaglutide and semaglutide on the endpoints analysed, a sample size determination was not performed. 

### 2.5. Statistical Analysis

All of the statistical analyses were performed using the Statistical Package for Social Sciences version 25 (SPSS, Inc., Chicago, IL, USA). Continuous variables were expressed as mean and standard deviation (SD) or median and interquartile range (IQR), when appropriate. The normal distribution of each variable was evaluated with the Kolmogorov-Smirnov test with Lilliefors correction. Differences in the values of the explanatory variables were compared by a *t*-test for paired samples in cases of normally distributed data. For non-normal distributed variables, a Wilcoxon signed-rank test was used. Categorical variables were expressed as percentage values and compared by a X^2^ test or Fisher exact test, when appropriate. Furthermore, the association of the relevant baseline clinical variables with the mean difference of LV GLS after 6 months of GLP-1RAs treatment was explored. For continuous baseline variables, the Pearson’s or Spearman’s correlation coefficient was used, when appropriate. Baseline categorical variables were explored by point-biserial correlation A *p* value < 0.05 was set for statistical significance.

## 3. Results

Between December 2019 and March 2020, 22 patients were enrolled for a mean follow-up of 150 ± 41 days. The baseline characteristics of the study population are listed in Table 1. The mean age of the sample was 65 ± 10 years, with a small prevalence of male sex (64%). All of the patients were at high or very high risk of ASCVD, due to the presence of several CV risk factors, in addition to DM2. Indeed, the vast majority of the patients had arterial hypertension (100%), dyslipidaemia (95%), family history of CV disease (91%), obesity (86%) and a smoking habit (68%). Furthermore, two patients were affected by coronary artery disease (CAD), with a previous myocardial infarction (MI), leading to a moderate LV systolic dysfunction in one of them. Comorbidities are not rare in this setting of patients and were present in almost all of the patients (95%). Some of the patients were already taking oral antidiabetic drugs such as metformin, while none of them were taking insulin. The echocardiographic data are summarized in Table 2. A mild LV concentric hypertrophy was highly represented [median interventricular septum was 12 (IQR 1) mm and mean LV mass 154 ± 22 g]. The median LVEF was 59% (IQR 7), suggesting that systolic dysfunction was rare. The right heart was within the normal range in all of the patients, whilst the diastolic function was mildly impaired [median E/A 0.75 (IQR 0.28) and mean E/E’ 7 ± 2]. No one had valvular diseases, except for minimum/mild tricuspid and mitral regurgitation. The mean LV GLS was slightly under the normal range (−18.0 ± 1.8%). The association between the baseline clinical data and the LV GLS variations from baseline to the 6-month follow-up was explored, and the results are listed in Table 3. No significant interactions emerged from the analysis. After six months of treatment with semaglutide or dulaglutide, we observed a clinical and statistically significant (mean difference −1.4 ± 1.1%; *p* < 0.001) improvement in the LV GLS, with a mean value tending towards the normal range (19.5 ± 2.1%) (Figure 2). No relevant changes were seen in the other echocardiographic parameters, with the absence of considerable drug side effects during the follow-up period.

## 4. Discussion

To the best of our knowledge, this is the first clinical study that has sought to evaluate the effect of GLP-1RAs semaglutide and dulaglutide on cardiac function by employing echocardiography and, in particular, STE in a cohort of consecutive DM2 patients with ASCVD or high or very high CV risk. The main results of the current analysis are that myocardial contraction, as assessed by GLS-STE, significantly improved after six months of GLP-1RAs semaglutide and dulaglutide therapy. 

Several CVOTs pointed out the cardioprotective role of these drugs [10,13,15,16,17]. The SUSTAIN-6, a preapproval safety CVOT, demonstrated a reduction in the rate of death from CV causes, non-fatal myocardial infarction and non-fatal stroke with semaglutide compared to the placebo after two years of follow-up [16]. In addition, the REWIND trial, which tested dulaglutide and the LEADER trial with liraglutide, showed a reduction in the MACE compared to the placebo [15,17]. According to this, the ESC guidelines recommend liraglutide, semaglutide or dulaglutide (class of recommendation I level of evidence A) in patients with DM2 and very high/high CV risk [18]. However, the exact cardioprotective mechanisms of GLP-1RAs still need to be clarified. 

Although there is a lack of a universally accepted definition, diabetic cardiomyopathy (DCM) is defined as the presence of ventricular dysfunction in absence of hypertension, CAD, congenital or valvular heart disease and may evolve towards overt heart failure [26,27,28]. Diastolic dysfunction with a preserved LVEF is considered the first marker of DCM [29]; however, some studies have observed that it is preceded by an earliest reduction in LV GLS [30,31,32]. This subclinical LV dysfunction observed in asymptomatic DM2 patients is common and associated with a long-term worse outcome, with a high rate of mortality and hospitalization [23]. The development of these subtle alterations in the LV function, however, is multifactorial. One suggested mechanism consists of insulin resistance as a cause of impaired cardiac metabolic flexibility. The reduced uptake of glucose from the cardiomyocytes may lead to a substrate shift towards increased fatty acids oxidation, causing lipotoxicity, and, therefore, reduced cardiac efficiency, as well as an increase in myocardial oxygen consumption and the production of ROS (Reactive Oxygen Species). Furthermore, hyperglycaemia itself may promote the development of AGEs (Advanced Glycated End-Products) stimulating collagen expression and accumulation, leading to myocardial fibrosis with an increase in myocardial stiffness and reduced cardiac compliance [28,33,34,35]. High glucose levels also cause Ca^2+^ mishandling and mitochondrial dysfunction, which may play a crucial role in the earliest stages of DCM [36,37].

Several preclinical studies on animal models and clinical studies have demonstrated that GLP-1 and its analogues exert several beneficial actions on the cardiovascular system [14,38,39,40,41,42,43,44]. On cardiomyocytes, these are mainly indirect and manifest through better glycaemic control, weight loss and blood pressure lowering [14,45,46,47]. However, GLP-1 receptors are highly expressed in atrial and vascular cells; therefore, GLP-1 Ras could exert, at least partialy, a direct action on the cardiac tissue [45,48]. The reduction in insulin resistance induced by GLP1-RAs may restore cardiac metabolic flexibility through the improvement of the insulin receptor, increasing glucose uptake and utilization [45,49]. After the infusion of GLP-1, Sokol et al. highlighted an improvement in the systolic function and functional status of patients with chronic heart failure [40]. On ischemic myocardium, GLP-1 RAs may prevent cardiomyocyte apoptosis and improve cardiac function via the AMPK/phosphoinositide 3-kinase (PI3K)–protein kinase B (Akt) pathway [50]. This antiapoptotic effect may explain the reduction in the infarct size and myocardial stunning in an in vivo rabbit model of myocardial ischemia/reperfusion with a single dose of parenteral fusion protein GLP-1-Tf (transferrin) [51]. These results suggest that GLP1-RAs may play a pivotal role in preventing the development of DCM, particularly in those with an early subclinical LV dysfunction. 

GLS-STE represents an innovative technique for the evaluation of myocardial function [21,22]. Numerous studies have shown its incremental diagnostic and prognostic value in multiple cardiovascular conditions, such as ischaemic heart disease [52] and cardiomyopathies [53]. Indeed, despite the normal values of LVEF, the evidence of the early impairment of LV GLS reveals a subclinical systolic dysfunction that may evolve towards clinically relevant systolic impairment and cardiac complications [23,54]. Previously, different echocardiographic techniques, including GLS-STE, were used for assessing the cardiac modifications derived by GLP1-RAs introduction. In fact, Lambadiari et al. found a modest increase in LV GLS after six months of therapy with liraglutide compared to metformin in subjects with newly diagnosed DM2 [55]. Our findings are in line with these preliminary results. However, as expected, the literature presents incongruent data on the topic. Bizino et al. pointed out, with the employment of cardiac magnetic resonance (CMR) in DM2 patients treated with liraglutide, a reduction in the LV filling pressure with a non-significant reduction in the systolic function after 26 weeks of treatment [56], and the same results were obtained by Yari et al. [57] The meta-analysis by Zhang et al. in a larger sample of 4790 subjects, highlights an increase in LVEF and a reduction in LVESV and E/e’ with GLP1-RAs compared to the placebo [20]. In the present study. we did not confirm these results because the LVEF remained nearly unchanged in the follow-up evaluation. Diastolic dysfunction with a preserved LVEF was considered another early sign of subclinical myocardial damage in diabetic patients [58]. Bizino et al. demonstrated in DM2 patients, with the use of CMR, that liraglutide reduces early LV diastolic filling and LV filling pressure, unloading the left ventricle, in a treatment interval of 26 weeks. Subsequently, these results were confirmed in a meta-analysis by Zhang et al., which highlighted a reduction in E/e’ after therapy [20]. Despite the fact that the positive effect on the diastolic function may be related to a direct improvement of GLP-1RAs treatment on LV cardiac remodelling, drug-induced weight loss may play a relevant role as an indirect mechanism [58]. Unexpectedly, in the present analysis, we did not find a relevant variation in the diastolic parameters; this was likely due to the small sample size. 

In general, because of the heterogeneity of the study designs and the different pharmacological properties, any head-to-head comparison between different GLP-1 RA analogues should be interpreted cautiously. Therefore, further studies in larger populations and with a longer follow-up duration are needed to better elucidate the effect of these drugs on cardiac function.

### Study Limitations

This study has some limitations. First, the reduced sample size led to weak statistical power. The identification of a difference between the baseline and after six months LV GLS requires further studies with a larger sample size to confirm these results. The second limitation is the absence of a control group and the lack of blinding, which may have led to bias in the evaluation of the echocardiographic parameters during the follow-up.

## 5. Conclusions

In the present study, we observed that six months of treatment with GLP-1 RAs dulaglutide or semaglutide leads to an improvement of LV GLS in subjects with DM2 with a high or very high risk for ASCVD or with ASCVD. Accordingly, these antidiabetic drugs should be preferred in this setting of subjects in order to prevent the development of DCM. Further studies on larger populations and with a longer duration of follow-up are warranted to confirm these preliminary results.

## Figures and Tables

**Figure 1 jcm-12-01586-f001:**
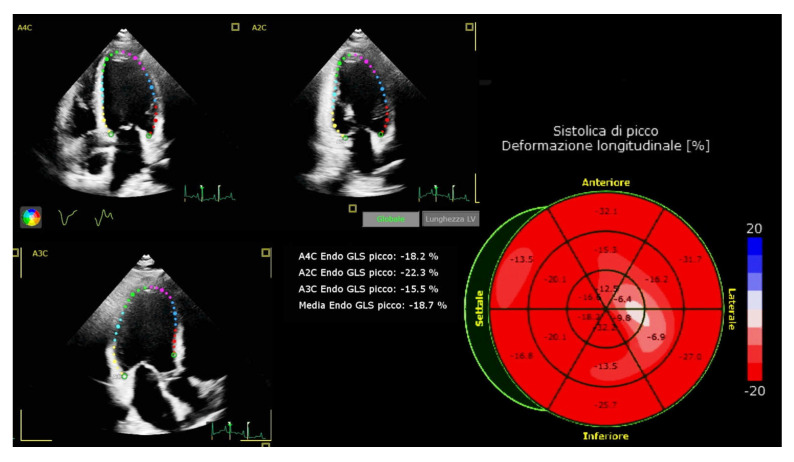
Left Ventricular Global Longitudinal Strain (LV GLS) with speckle tracking echocardiography (STE) is a helpful tool able to find out a subclinical LV dysfunction in an otherwise normal heart, which may evolve towards diabetic cardiomyopathy. Peak GLS is a negative number and a value in the range of −20% is considered normal.

**Figure 2 jcm-12-01586-f002:**
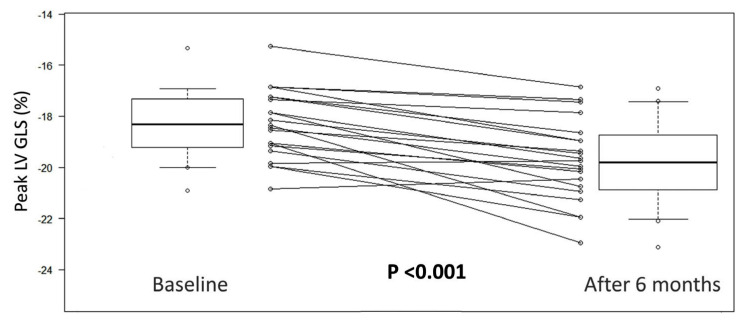
Peak Left Ventricular Global Longitudinal Strain (LV GLS) at baseline and after 6 months of treatment. The boxplots and the lines graph show a statistically significant (*p* < 0.001) improvement of LV GLS after 6 months of treatment with GLP1-RAs.

**Table 1 jcm-12-01586-t001:** Baseline characteristics of the study population. BSA: Body Surface Area; BMI: Body Mass Index; SBP: Systolic Blood Pressure; DBP: Diastolic Blood Pressure; CAD: Coronary Artery Disease; MI: Myocardial Infarction; COPD: Chronic Obstructive Pulmonary Disease; CKD: Chronic Kidney Disease; PAD: Peripheral Artery Disease; eGFR: estimated Glomerular Filtration Rate; ALT: Alanine-amino Transferase.

Variable	All Patients (*n* = 22)
Demographic and anthropometric data	
Age, years, mean ± SD	65 ± 10
Sex, male, *n* (%)	14 (64)
BSA, m^2^, mean ± sd	2.03 ± 0.21
BMI, m^2^/Kg	33.9 ± 5.1
Clinical data	
SBP, mmHg, mean ± sd	133 ± 13
DBP, mmHg, mean ± sd	79 ± 10
Heart rate, bpm	71 ± 10
Cardiovascular risk factors other than DMT2	
Hypertension, yes, *n* (%)	22 (100)
Dyslipidaemia, yes, *n* (%)	21 (95)
Smoking habit, yes, *n* (%)	15 (68)
Obesity, yes, *n* (%)	19 (86)
Family history of cardiovascular disease, yes, *n* (%)	20 (91)
Comorbidities	
CAD, yes, *n* (%)	2 (9)
Previous IM, yes, *n* (%)	2 (9)
COPD, yes, *n* (%)	7 (32)
CKD, yes, *n* (%)	2 (9)
PAD, yes, *n* (%)	3 (14)
Thyroid disorders, yes, *n* (%)	4 (18)
Medical therapy	
ACE-I/ARB, yes, *n* (%)	18 (82)
Beta-blockers, yes, *n* (%)	10 (45)
Diuretics, yes, *n* (%)	8 (36)
Anticoagulant therapy, yes, *n* (%)	2 (9)
Anti-thrombotic agents, yes, *n* (%)	11 (50)
Statins, yes, *n* (%)	15 (68)
Metformin, yes, *n* (%)	21 (95)
Insulin, yes, *n* (%)	0 (0)
Laboratory testing	
Creatinine, mg/dL, median [IQR]	0.87 [0,31]
eGFR, mL/min/1.73 m^2^, median [IQR]	83 [26]
Uric acid, mg/dL, median [IQR]	5.6 [2,9]
Plasma sodium, mEq/L, mean ± sd	139.5 ± 1.8
Plasma potassium, mEq/L, median [IQR]	4.2 [0,4]
Haemoglobin, g/dL, mean ± sd	13.6 ± 1.4
Glycated haemoglobin, mMol/Mol, median [IQR]	49 [16]
Fasting plasma glucose, mg/dL, median [IQR]	109 [25]
Total bilirubin, mg/dL, median [IQR]	0.6 [0,2]
ALT, mg/dL, median [IQR]	25 [14]

**Table 2 jcm-12-01586-t002:** Echocardiographic data at baseline and after 6 months of GLP-1RA treatment. LV: Left Ventricle; PW: Posterior Wall; EDD: End Diastolic Diameter; EDV: End Diastolic Volume; EF: Ejection Fraction; ESV: End Systolic Volume; TAPSE: Tricuspid Annular Plane Systolic Excursion; LA: Left Atrium; APD: Antero-posterior Diameter; RVOT: Right Ventricle Outflow Tract; RV: Right Ventricle; EDA: End Diastolic Area; ESA: End Systolic Area; FAC: Fractional Area Change; DT: Deceleration Time; SPAP: Systolic Pulmonary Artery Pressure; IVC: Inferior Vena Cava; GLS: Global Longitudinal Strain.

Variable	Baseline (*n* = 22)	After 6 Months (*n* = 22)	Difference (*n* = 22)	*p* Value
LV septum, mm, median [IQR]	12 [1]	12 [1]	0 [0]	0.257
LV PW, mm, median [IQR]	11 [1]	11 [1]	0 [0]	0.206
LV EDD, mm, mean ± sd	47 ± 5	46 ± 5	1 ± 3	0.158
LV Mass, g, mean ± sd	154 ± 22	153 ± 21	1 ± 2	0.560
LV EDV, mL, median [IQR]	98 [24]	98 [23]	0 [11]	0.835
LV ESV, mL, median [IQR]	40 [19]	41 [16]	0 [6]	0.615
LV EF, %, median [IQR]	59 [7]	58 [8]	1 [6]	0.776
LA APD, mm, median [IQR]	41 [3]	40 [3]	0 [2]	0.273
LA Area, cm^2^, median [IQR]	17 [5]	17 [4]	1 [2]	0.024
LA Volume, mL, mean ± sd	50 ± 13	48 ± 13	2 ± 7	0.278
RVOT, mm, mean ± sd	31 ± 4	32 ± 3	−1 ± 10	0.294
RV basal, median [IQR]	35 [4]	36 [4]	−1 [2]	0.059
RV EDA, cm^2^, mean ± sd	15 ± 2	15 ± 2	0 ±2	0.102
RV ESA, cm^2^, mean ± sd	8 ± 1	7 ± 1	1 ± 1	0.005
RV FAC, %, mean ± sd	46 ± 6	48 ± 8	−2 ± 8	0.172
E wave, cm/s, mean ± sd	64 ± 13	64 ± 13	0 ± 16	0.863
A wave, cm/s, mean ± sd	85 ± 18	83 ± 14	2 ± 13	0.545
E/A, median [IQR]	0.75 [0,28]	0.78 [0,19]	−0.02 [0,15]	0.433
DT, msec, mean ± sd	175 ± 45	166 ± 39	9 ± 34	0.236
S’ septal, cm/s, mean ± sd	8 ± 2	8 ± 1	0 ± 2	0.390
E’ septal, cm/s, mean ± sd	9 ± 2	8 ± 2	1 ± 2	0.280
A’ septal, cm/s, mean ± sd	10 ± 2	11 ± 2	−1 ± 2	0.196
S’ mitral, cm/s, mean ± sd	9 ± 2	9 ± 2	0 ± 2	0.853
E’ mitral, cm/s, mean ± sd	10 ± 3	10 ± 3	0 ± 2	0.741
A’ mitral, cm/s, mean ± sd	11 ± 3	11 ± 3	0 ± 2	0.343
E/E’, mean ± sd	7 ± 2	7 ± 2	0 ± 2	0.589
S’ tricuspid, cm/s, median [IQR]	12 [2]	12 [3]	0 [2]	0.751
E’ tricuspid, cm/s, median [IQR]	11 [2]	11 [2]	0 [1]	0.842
A’ tricuspid, cm/s, median [IQR]	9 [5]	13 [6]	−1 [5]	0.003
TAPSE, mm, mean ± sd	21 ± 2	22 ± 2	−1 ± 2	0.392
SPAP, mmHg, median [IQR]	26 [10]	26 [9]	0 [4]	0.362
IVC diameter, mm, median [IQR]	18 [2]	18 [3]	0 [1]	0.361
IVC collapse, no, *n* (%)	22 (100)	21 (95)	-	nv
LV GLS, %, mean ± sd	−18.0 ± 1.8	−19.5 ± 2.1	−1.4 ± 1.1	<0.001

**Table 3 jcm-12-01586-t003:** Correlations between baseline clinical data and variation of LV GLS after 6-months of treatment. BMI: Body Mass Index; SBP: Systolic Blood Pressure; DBP: Diastolic Blood Pressure.

Variabile	Coefficient	*p* Value
Age	0.239	0.285
Sex	−0.045	0.843
BMI	0.364	0.096
Smoking habit	−0.431	0.045
SBP	0.059	0.796
DBP	−0.237	0.287
Heart Rate	0.054	0.817
Therapy with metformin	−0.209	0.351
Gylicated haemoglobin	0.019	0.933
Fasting plasma glucose	−0.007	0.975

## Data Availability

The data that support the findings of this study are available from the corresponding author upon reasonable request.

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
