# Peer review of "Improvement of Left Ventricular Global Longitudinal Strain after 6-Month Therapy with GLP-1RAs Semaglutide and Dulaglutide in Type 2 Diabetes Mellitus: A Pilot Study"

_jcm, 2023, doi:10.3390/jcm12041586_

Round 1
Reviewer 1 Report
Review report
In this current report, P. Basiel and colleagues conducted an observational, perspective, monocentric study in a cohort of consecutive patients with DM2 to evaluate the possible effects of GLP1-R agonists on cardiac function measured by Speckle Tracking Echocardiography.
Overall, the work is valuable, the methods are clearly presented, and the results are mostly well-supported. This reviewer has some issues regarding the English language editing, statistics, and some contradictory findings with the literature (e.g. the lack of diastolic dysfunction improvement). Most importantly, some association analyses should have been performed, as the data was available.
Major comments
It would be valuable to perform association studies – how the measured echo parameters are influenced by the patients data, e.g. smoking habit, metformin use, HBA1c, etc.
Minor comments and typos:
Extensive English review is required (mostly: disfunction –dysfunction, etc…). Please, carefully edit the manuscript.
Abstract/
Row 23 GLP1 Ras – improper abbreviation (RAs)
Row 26-27 rephrase sentence to improve clarity
Introduction/
+please, provide a short subsection to revise the STE echo technique
42 – glycemia
Methods/
2.3. Echo – please, clarify if foreshortening was evaluated before recording the traces to measure strain
Row 126 – statistics – please explain why the Kolmogorov normality test was chosen
Results/
Row 146 – disfunction –dysfunction
Also, it would be valuable to perform circumferential strain analysis, and segmental strain or time-to-peak (TPK) analyses to better assess the cardiac strain in association with the GLP-1 treatment.
Discussion/
Please, discuss the contradictory findings with the literature in more details, e.g. why diastolic dysfunction and other systolic parameters failed to improve after the treatments.
Reviewer 2 Report
This observational study is of high clinical interest. The methodology is accurate and the results are presented with clarity. The discussion is pertinent and informative. The main limitation is the small number of patients included in the study, a point highlighted by the authors. I have fourspecific comments:
1) Please include information on the patient cohort in the abstract
2) Please include also calcium channel antagonists in the list of medication
3) Please provide a power analysis for two or three main echocardiographic parameters
4) Please correct some trivial typographic errors: e.g. line 27: 'associated with' line 122: 'statistical', Table 1 capitalize sd-> SD
Reviewer 3 Report
the author must specify the recording of blood pressure during the echo procedure and the device used. this is a major confounding factor in this study
Round 2
Reviewer 1 Report
Thank you for addressing my issues.
No further comments.